# Continuous Glucose Monitoring (CGM) in Sports—A Comparison between a CGM Device and Lab-Based Glucose Analyser under Resting and Exercising Conditions in Athletes

**DOI:** 10.3390/ijerph20156440

**Published:** 2023-07-25

**Authors:** Helen Bauhaus, Pinar Erdogan, Hans Braun, Mario Thevis

**Affiliations:** 1Institute of Biochemistry, German Sport University Cologne, 50933 Cologne, Germany; 2German Research Centre of Elite Sports, German Sport University Cologne, 50933 Cologne, Germany; h.braun@dshs-koeln.de; 3Manfred Donike Institute for Doping Analysis, 50933 Cologne, Germany; 4Centre for Preventive Doping Research, German Sport University Cologne, 50933 Cologne, Germany

**Keywords:** continuous glucose monitoring, application in sports, carbohydrate management, active subjects, validation

## Abstract

The objective of this pilot study was to compare glucose concentrations in capillary blood (CB) samples analysed in a laboratory by a validated method and glucose concentrations measured in the interstitial fluid (ISF) by continuous glucose monitoring (CGM) under different physical activity levels in a postprandial state in healthy athletes without diabetes. As a physiological shift occurs between glucose concentration from the CB into the ISF, the applicability of CGM in sports, especially during exercise, as well as the comparability of CB and ISF data necessitate an in-depth assessment. Ten subjects (26 ± 4 years, 67 ± 11 kg bodyweight (BW), 11 ± 3 h) were included in the study. Within 14 days, they underwent six tests consisting of (a) two tests resting fasted (HC_Rest/Fast and LC_Rest/Fast), (b) two tests resting with intake of 1 g glucose/kg BW (HC_Rest/Glc and LC_Rest/Glc), (c) running for 60 min at moderate (ModExerc/Glc), and (d) high intensity after intake of 1 g glucose/kg BW (IntExerc/Glc). Data were collected in the morning, following a standardised dinner before test day. Sensor-based glucose concentrations were compared to those determined from capillary blood samples collected at the time of sensor-based analyses and subjected to laboratory glucose measurements. Pearson’s r correlation coefficient was highest for Rest/Glc (0.92, *p* < 0.001) compared to Rest/Fast (0.45, *p* < 0.001), ModExerc/Glc (0.60, *p* < 0.001) and IntExerc/Glc (0.70, *p* < 0.001). Mean absolute relative deviation (MARD) and standard deviation (SD) was smallest for resting fasted and similar between all other conditions (Rest/Fast: 8 ± 6%, Rest/Glc: 17 ± 12%, ModExerc/Glc: 22 ± 24%, IntExerc/Glc: 18 ± 17%). However, Bland–Altman plot analysis showed a higher range between lower and upper limits of agreement (95% confidence interval) of paired data under exercising compared to resting conditions. Under resting fasted conditions, both methods produce similar outcomes. Under resting postprandial and exercising conditions, respectively, there are differences between both methods. Based on the results of this study, the application of CGM in healthy athletes is not recommended without concomitant nutritional or medical advice.

## 1. Introduction

It is generally known that substrate oxidation changes when increasing the intensity of exercise, with the increasing contribution of muscle glycogen and blood glucose to energy provision [1,2,3]. Therefore, regular carbohydrate (CHO) intake during exercise, which should be adapted to the overall duration of the exercise, is recommended [4,5,6]. While free blood glucose concentration is no proper indicator of CHO availability, maintaining adequate blood glucose levels during exercise is assumed to reflect a successful CHO intake strategy during exercise [7]. Sensor-based continuous glucose monitoring (CGM) is a minimally invasive technology that allows for the recognition of glucose flux and is known as a therapeutic application for individuals managing diabetes. Its utility was demonstrated in the context of glycaemic control, reduced periods of hypoglycaemia, and reduced glycolated haemoglobin (HbA1c) concentrations [8,9].

In order to achieve the best possible athletic performance, the use of innovative technology to support the monitoring of critical parameters in training periods and in-competition strategies has long reached professional sports. Therefore, athletes’ training plans and training sessions are accompanied by non-invasive and continuous measurements of parameters such as heart rate (HR), global positioning system (GPS), accelerometry or core temperature [10]. Since continuous glucose monitoring (CGM) devices have become widely accessible, a considerable interest was recognised amongst athletes of endurance-based sport disciplines, aiming at optimizing physiological variables, e.g., by tracing blood glucose levels. CGM devices in sports are promoted to enable CHO intake control during physical activity to enhance energy supply and, therefore, to improve athletic performance. For instance, Jan Frodeno (Triathlon, GER), Eluid Kipchoge (Marathon, KEN), Kristian Blumenfelt (Triathlon, NOR) and Sophie Power (Ultrarunning, UK) are prominent examples of endurance athletes living without diabetes but using sensor-based glucose concentration analysis. Different from conventional glucose testing, which is commonly conducted with capillary blood (CB) and which was applied in recent performance studies that produced recommendations for CHO intake, sensors measure glucose subcutaneously in the interstitial fluid (ISF). Recent studies on diabetics report a lag time (=delayed onset of CGM curve compared to CB curve) of 5 to 25 min depending on physiological changes during exercise, alterations in blood flow rate, body temperature and body acidity [8,11,12,13]. As also healthy people are advised to avoid hypoglycemia during exercise in order to delay fatigue, the occurrence of a lag time probably evokes a delayed reaction and experiencing hypoglycemia. Despite the increasing interest and application in athletes, data on healthy, active subjects are missing [7]. The authors hypothesise that there will be a difference between CB and ISF in healthy subjects, just as previously seen in people with diabetes and due to the physiological shift between ISF and CB glucose. Therefore, the objective of this exploratory pilot study was to compare glucose concentration in capillary blood (CB) samples analysed using a validated method and glucose concentration measured in the ISF by CGM under different physical activity levels and different nutritional statuses in athletes in order to evolve the applicability of CGM as a useful tool during exercise. 

## 2. Methods

### 2.1. Subjects

Healthy, non-smoking subjects between 18 and 35 years with an average training volume of >8 h/week were included in the study. For data analysis, data of 10 athletic subjects (4 females, 6 males) were considered involved in the following sports at a competitive level: crossfit (*n* = 1), running (*n* = 5), triathlon (*n* = 3), and cycling (*n* = 1). At the beginning of the study, subjects were 26 ± 4 years, had a bodyweight (BW) of 67 ± 11 kg and a training volume of 11 ± 3 h. All of them were following a training plan that they were given by their coach. The study took place in-season; therefore, the training, study protocol, and work/studies of the subjects needed to be coordinated. The subjects were recruited from their related sports club. Subjects provided written informed consent before participation. The study was approved based on the Declaration of Helsinki by the university’s local ethics committee (159/2021) before the start of data collection.

### 2.2. Glucose Monitoring Devices

Glucose concentration was monitored using two different devices (a) measuring glucose concentration from CB samples and (b) in the ISF, respectively. Both devices are based on an enzymatic–amperometric measurement technique. (1) In the first reaction, the C1-atom of beta-D-glucose present in the sample is oxidised into D-glucono-delta-lactone by glucose oxidase (GOD). (2) At the same time, GOD-bound FAD is reduced to GOD-FADH_2_. (3) D-glucono-delta-lactone can be further hydrolysed into gluconic acid. (4) In a side reaction of (2), GOD-FAD is oxidised to GOD-FADH_2_ by donating electrons to oxygen, which leads to the formation of hydrogen peroxide. The amperometric potential of hydrogen peroxide or oxygen, respectively, is detected by an electrode. Because of the stochiometric relation of glucose and hydrogen peroxide or oxygen, respectively, the signal can be translated into a glucose concentration [14].

(a)Capillary Blood Glucose Analysis

CB samples were analysed using Biosen C-Line (EKF diagnostics Holding, Cardiff, UK). Biosen C-Line measures glucose and lactate in human blood, plasma, and serum samples. The manufacturer reports a precision of ≤1.5% (at 12 mmol). Biosen C-Line was calibrated against a 12 mmol/L glucose solution standard each morning before the first sample array was measured. The amperometric signal of the standard forms the basis for the calculation of unknown glucose concentrations. Capillary blood samples were taken at the earlobe and stored in end-to-end capillary tubes containing a haemolysis solution (EKF diagnostics Holding, Cardiff, UK) to avoid metabolic degradation of glucose. Under resting conditions, subjects were asked to gently massage their earlobe for ca. 20 s before the next sampling to guarantee blood flow. Glucose values are measured in the manner described previously. The validity and reliability of the Biosen C-Line were confirmed previously [15,16]. For this study, Biosen C-Line was used as the reference method.

(b)Interstitial Fluid Glucose Analysis

ISF glucose concentration was measured using a Libre Sense Glucose Biosensor for Sport (Abbott Laboratories, Chicago, IL, USA). For accessing crude minute-per-minute glucose concentration data, software by SuperSapiens (TT1 Products Inc., Atlanta, GA, USA) was used. Glucose sensors are equipped with a 4 mm needle that is implanted in the dermis (thickness of 1–4 mm), which lies between the epidermis (thickness of 75–150 µm) and subcutaneous tissue (1–20 mm) [11]. The needle carries the enzyme glucose oxidase and reacts oxygen-dependent in the manner described above [17,18]. According to the manufacturer’s guidelines, no calibration with blood glucose is needed for the sensors used in this study. Validity and reliability tests are being performed at the moment, but no data have been published yet. Common body sites for sensor location are periumbilical or at the back of the forearm. In line with the manufacturer’s recommendation, in this study, the sensor was applied at the back of the upper arm of the subjects. For continuous data generation, a Bluetooth connection to an NFC-enabled phone is necessary. Libre Sense Glucose Biosensor for Sport was used as the comparator method.

### 2.3. Test Protocol

Subjects underwent six tests at the laboratory, which are presented in the current article, plus three tests focusing on the ISF glucose response to three different foods matched for a content of 20 g CHO, which will not be covered further here. All tests were conducted within a time frame of 14 days, as sensor durability is limited. The tests were aligned with the subjects’ training and working/study schedule and were therefore not randomised but individually planned to reach the highest possible compliance and facilitate subject recruitment. As recommended by the manufacturer, no tests were conducted within the first 24 h due to sensor calibration. Subjects arrived at the laboratory in the morning after a 9 h overnight fast. Dinner before the overnight fast was standardised according to the test protocol. Dinner recipes were isocaloric but differed in macronutrient distribution (Table 1). Depending on the test protocol, dinner was low, moderate, or high in CHO, respectively (Figure 1). The six different test protocols were (1) resting fasted with a high CHO dinner before overnight fast (HC_Rest/Fast), (2) resting fasted with a low CHO dinner before the overnight fast (LC_Rest/Fast), (3) resting after intake of 1 g glucose/kg BW with a high CHO dinner before overnight fast (HC_Rest/Glc), (4) resting after intake of 1 g glucose/kg BW with a low CHO dinner before overnight fast (LC_Rest/Glc), (5) moderate running (65% HF_max_) for one hour after intake of 1 g glucose/kg BW with a moderate CHO dinner before overnight fast (ModExerc/Glc) and (6) intensive running (85% HF_max_) for one hour after intake of 1 g glucose/kg BW with a moderate CHO dinner before overnight fast (IntExerc/Glc) (Figure 1). Different CHO loads before the resting fasted and resting postprandial protocols were applied as the authors wanted to see a possible influence of the CHO load of the dinner on glucose values in the different media. Before exercising conditions, the subjects’ dinner was moderate in CHO (1.25 g/kg BW) to guarantee sufficient CHO intake before one hour of exercise the next morning. Due to COVID-19 restrictions, lab-based performance diagnostics for the determination of HR_max_ could not be performed. Instead, the age-derived equation by Fox et al. (1968) was used to estimate HR_max_ by calculating the difference of 220 minus the subject’s age [19]. By that, subjects’ 65% and 85% HR_max_ reflecting moderate and intensive exercise load, respectively, was calculated. This method was considered the most practical possible under given circumstances. As CB and ISF data of the same person were matched and their related glucose kinetics should be analysed in the first place, this method is not considered a full limitation of the study.

### 2.4. Test Procedure

Before the test started, subjects were instructed to sit relaxed for 30 min. CB samples for HC_Rest/Fast and LC_Rest/Fast were taken at baseline (BL) and after 5, 10, 15, 30, 45, 60, 75, 90, 105 and 120 min. For HC_Rest/Glc and LC_Rest/Glc, subjects ingested 1 g Glc/kg BW solved in 500 mL of water, followed by resting for 120 min. CB samples were taken at BL, right after glucose solution ingestion, after 5, 10, 15, 30, 45, 60, 75, 90, 105 and 120 min. For ModExerc/Glc and IntExerc/Glc, after BL measurement, subjects ingested 1 g Glc/kg BW, followed by resting for 30 min first and then starting the exercise program that consisted of running for 60 min at a constant moderate (ModExerc/Glc) and intensive (IntExerc/Glc) load, respectively. CB samples were taken at BL, right after glucose solution ingestions, after 10, 20, 30, 40, 50, 60, 70, 80, and 90 min. Running protocols were separated by at least 48 h. Running tests were performed at the university’s 400 m-tartan track in January and February 2022. Sampling was always performed in the same place. One researcher per subject was present to guarantee time accuracy in CB sampling. All subjects received their individual test schedule and BW-related recipes as a pdf-document.

### 2.5. Data Access and Statistical Analysis

The analysis included 519 data pairs (HC_Rest/Fast: 100; LC_Rest/Fast: 64; HC_Rest/Glc: 87; LC_Rest/Glc: 108; ModExerc/Glc: 94, IntExerc/Glc: 82). For analysis, 91% (HC_Rest/Fast), 58% (LC_Rest/Fast), 73% (HC_Rest/Glc), 90% (LC_Rest/Glc), 85% (ModExerc/Glc) and 75% (IntExerc/Glc) of possible total paired data were available. Lacking data pairs were mainly due to sensor errors or Bluetooth connection failure. CB glucose data were accessed via Biosen C-Line printout after measurement of each subject’s array. The time of CB sample collection was documented during data collection to match continuous glucose data. Statistics were performed using Excel 2019 (Microsoft Corp., Redmond, WA, USA) and Statistical Package for Social Sciences 28 (IBM SPSS Statistics, Chicago, IL, USA). The statistical difference of BL, maximum value (PEAK), area under the curve (AUC) between HC_Rest/Fast and LC_Rest/Fast, such as HC_Rest/Glc and LC_Rest/Glc, was analysed using paired parametric or non-parametric tests. A parametric test was performed after the normal distribution of residuals was confirmed via graphical analysis (histogram and QQ-plot) and Shapiro–Wilk test. Wilcoxon test was performed for non-parametric analysis. For the comparison of ISF and CB glucose values, data were analysed via linear regression analysis and Pearson’s correlation coefficient, systematic measurement difference (bias) [20], and mean absolute relative deviation (MARD). For systematic measurement difference, the mean of paired ISF and CB data was plotted against their difference. The difference between ISF and CB data was analysed for normal distribution by graphical analysis (histogram and QQ-plot) and Shapiro–Wilk test. MARD was calculated by |(GLC_ISF_ − GLC_CB_)|/GLC_CB_ (%) [21]. For all statistical tests, alpha was set at 5%. The effect size was calculated post hoc using Cohen’s d = $|\bar{x}|/({sd}_{1}+{sd}_{2})/2$ using G*Power. [22,23,24]. Study design, data analysis, reporting of results and interpretation were conducted under consideration of the Checklist for statistical Assessment of Medical Papers (CHAMP) [25].

## 3. Results

### 3.1. Resting Fasted Condition

For the residuals of BL_CB_, BL_ISF_, AUC_CB_, and AUC_ISF_, normal distribution was confirmed. Parametric tests for BL_CB_ and BL_ISF_, respectively, did not reveal any statistical differences between HC and LC dinner. Parametric tests of AUC_CB_ and AUC_ISF_, respectively, showed a significant difference for AUC_ISF_ (t(3) = 4.683, 95%CI [549;2879], *p* < 0.018, d = 1.62) between HC and LC. Therefore, data of the test protocols HC_Rest/Fast and LC_Rest/Fast were analysed separately. As glucose fluctuations under resting and fasted conditions are low, detailed data of parametric and non-parametric test results such as MARD calculation, glucose curves reflecting mean ± standard deviation and Bland–Altman plot can be found in Appendix A.

### 3.2. Resting Postprandial Condition

For the residuals of BL_CB_, BL_ISF_, PEAK_CB_, PEAK_ISF_, and AUC_ISF_, normal distribution was confirmed. Parametric tests for BL and PEAK, such as non-parametric tests for AUC, did not reveal any statistical differences between HC and LC dinner (detailed data of parametric and non-parametric test results in Appendix A).

### 3.3. Resting Postprandial Condition—High Carbohydrate Dinner before Test Morning

Mean glucose concentrations, including SD and regression analysis, are displayed in Figure 2(a.i,a.ii). There is a positive linear correlation (r = 0.88, *p* < 0.001) between CB and ISF glucose concentration measurements. Systematic measurement difference shows 95% limits of agreement between −13 and 52 mg/dL (∆LOA 65 mg/dL) [20]. The mean difference between CB and ISF glucose values is 20 mg/dL (Figure 3a). MARD ± SD is 17 ± 10%. The dispersion of absolute relative deviation (ARD), as recommended by the Food and Drug Administration, is displayed in Figure 4 [26].

### 3.4. Resting Postprandial Condition—Low Carbohydrate Dinner before Test Morning

Mean glucose concentrations, including SD and regression analysis, are displayed in Figure 2(b.i,b.ii). There is a positive linear correlation (r = 0.9, *p* < 0.001) between CB and ISF glucose concentration measurements. Systematic measurement difference shows 95% limits of agreement between −7 and 47 mg/dL (∆LOA 54 mg/dL) [20]. The mean difference between CB and ISF glucose values is 20 mg/dL (Figure 3b). MARD ± SD is 17 ± 9%. The dispersion of absolute relative deviation (ARD), as recommended by the Food and Drug Administration, is displayed in Figure 4 [26].

### 3.5. Exercise at 65% HR_max_

Mean glucose concentrations, including SD and regression analysis, are displayed in Figure 2(c.i,c.ii). There is a positive linear correlation (r = 0.60, *p* < 0.001) between CB and ISF glucose concentration measurements. Systematic measurement difference shows 95% limits of agreement between −58 and 67 mg/dL (∆LOA 125 mg/dL) [20]. The mean difference between CB and ISF glucose values is 4 mg/dL (Figure 3c). MARD ± SD is 22 ± 24%. The dispersion of ARD is displayed in Figure 4.

### 3.6. Exercise at 85% HR_max_

Mean glucose concentrations, including SD and regression analysis, are displayed in Figure 2(d.i,d.ii). There is a positive linear correlation (r = 0.70, *p* < 0.001) between CB and ISF glucose concentration measurements. Systematic measurement difference shows 95% limits of agreement between −48 and 52 mg/dL (∆LOA 100 mg/dL) [20]. The mean difference between CB and ISF glucose values is 2 mg/dL (Figure 3d). MARD ± SD is 18 ± 17%. The dispersion of ARD is displayed in Figure 4.

## 4. Discussion

### 4.1. Impact of Physical Activity

To our knowledge, to date, there has been only one study published comparing ISF and CB data systematically in subjects without diabetes investigating different protocols, including the assessment of validity postprandially after different breakfasts, pre-, during and post-exercise [27]. In our study, systematic measurement difference shows a smaller mean bias under exercising compared to resting conditions (ModExerc/Glc: 4 mg/dL; IntExerc/Glc: 2 mg/dL vs. HC_Rest/Glc: 20 mg/dL; LC_Rest/Glc: 20 mg/dL). However, their 95% CI is higher under exercising conditions compared to resting conditions (∆LOA: ModExerc/Glc: 125 mg/dL; IntExerc/Glc: 102 mg/dL vs. HC_Rest/Glc: 65 mg/dL; LC_Rest/Glc: 55 mg/dL). Clavel et al. (2022) compared the CGM system (Freestyle Libre, Abbott, France) with a finger prick system (FreeStyle Optimum, Abbott, France) [27]. The FreeStyle Optimum is a self-monitoring blood glucose device, which is different from our comparator lab-based device. Eight subjects received standardised isocaloric breakfasts with different macronutrient distributions, which were either CHO-loaded or protein and fat loaded. Irrespectively of breakfast, authors reported a mean bias of −2.99 mg/dL and ∆LAO of 58.36 mg/dL within 60 min post-breakfast, a mean bias of −1.67 mg/dL and a ∆LAO of 36.02 within the 60 min after the post-breakfast period, that the authors claimed as pre-exercise. The exercise protocol started with a 10 min low-intensity run that was followed by high-intensity intermittent training, which was different from ours. The mean bias during 40 min of exercise was 12.25 mg/dL (∆LOA [45.60 mg/dL]). Within 30 min post-exercise, the mean bias was 4.18 mg/dL (∆LOA [58.8 mg/dL]). The findings by Clavel et al. (2022) were different from our findings, revealing the highest mean bias during exercising protocols and a similar ∆LAO between all conditions, concluding that accuracy is negatively influenced by physical activity [27]. However, we found a lower mean bias during exercise but a higher ∆LOA indicating a higher variability of ISF and CB glucose values. Studies that were performed on subjects with type 1 diabetes confirm our findings and the findings of Clavel et al. (2022), showing a higher deviation of ISF and CB values under exercising conditions compared to resting and/or fasted conditions [28,29,30,31,32,33]. Moser et al. (2018) assume that physiological changes during exercise, such as alterations in blood flow rate, body temperature and body acidity, can theoretically have an impact on interstitial glucose-sensing accuracy [34].

### 4.2. Impact of CHO Intake before Testing

Based on our findings showing a significant difference under resting fasted conditions related to the CHO load of the dinner at least 9 h before test morning, we assume changes in glucose flux from CB into the ISF, dependent on glucose availability. As some recommend arterial or arterialised venous blood glucose as the gold standard for glucose measurement [35], Siegmund et al. (2017) suggest that glucose measured in the ISF might be the more relevant physiological parameter than blood glucose values, as the ISF might be more sensitive in relation to changes in glucose concentrations than blood. Siegmund et al. (2017) further hypothesise that ISF glucose concentration might better reflect glucose status than CB does due to homeostatic regulation of glucose concentration in blood [12]. There are several studies trying to estimate glucose flux in healthy people [36] and people living with diabetes [37,38] under resting conditions. Taken from our and previous studies, varying glucose flux from CB into the ISF due to chronic or acute nutritional status and physical activity is one major factor influencing the evaluation of CGM devices when compared to blood glucose analysers [13,30,33,39]. Additionally, different body sites of sensor application show slight but no statistical differences [40,41]. Taken from our findings that show a different reaction of the CGM device during resting and fasted conditions between the different CHO loads of the dinner before test day, we want to encourage further studies to consider nutritional status and overall CHO intake. As most studies were performed on subjects with diabetes controlling for glycemia or insulinemia in order to keep subjects away from a serious hypo- or hyperglycaemic event, more data on healthy subjects and athletes, respectively, are needed to further assess the applicability in sports.

### 4.3. Conclusion for Practical Application

As people living with diabetes are using CGM systems in order to manage their CHO intake and insulin doses, the correct handling of CGM devices and data is essential. Therefore, the application of CGM systems in patients with diabetes is accompanied by medical and/or nutritional experts’ advice. People without diabetes, including athletes, are not necessarily professionally advised before starting the use of CGM devices as they are freely available on the market. The fact that the sensors measure glucose in the ISF and not in the CB requires consideration in data interpretation and, consequently, respective action by athletes concerning their nutrition. Depending on how well blood glucose is reflected in the interstitium, the use of CGM may or may not support efforts to avoid hypoglycaemia during exercise. Basu et al. (2013) investigated glucose flux from the blood into the interstitium in healthy subjects by labelled glucose and observed a physiological lag time of 5–6 min under resting conditions [36]. This raises the question of how ISF glucose values are potentially affected during physical activity. Therefore, in this study, blood glucose concentrations were compared with those measured in the interstitium during different standardised tests. Especially under exercising conditions, our results indicate that referring to ISF glucose data can lead to CHO intake mismanagement and is, therefore, not recommended for healthy athletes. If the accompanied use of digital applications connected to the CGM sensor can replace professional advice and educate the user on how to interpret data and how to react under different circumstances is a question for future research. Additionally, as stated by Moser et al. (2018), CGM devices pose a risk for skin irritation [42,43], sleep disruption due to the discomfort of wearing the sensor [44] and an overload of information for people with diabetes and cannot be excluded for application by people without diabetes. While applying the sensor at the back arm or any other body site (periumbilical or at the upper leg), an insight into the software still shows longer periods of data lacks, which is also indicated by the occasionally low percentage of paired data that were available for analysis (e.g., resting fasted condition with low CHO dinner before overnight fast: 58% of possible paired data). Additionally, sensors tended to show errors that required the replacement of the sensor with a new one. Especially during swimming, sensors seem to have problems with minute-by-minute data production. Further, some subjects reported that sensors were torn off during or after their swimming session. On the other hand, CGM devices pose a chance to raise athletes’ awareness of increasing their CHO intake as several studies found an insufficient daily carbohydrate intake, as indicated by a meta-analysis by Steffl et al. (2019) [45]. Spronk et al. (2014) assume a lack of nutritional knowledge to be one factor for insufficient nutritional status [46]. Additionally, yet unpublished data show an insufficient CHO intake four hours before, during and within two hours after exercise. If the application of CGM devices can support nutritional consultation as another tool in the practitioner’s toolbox, it should be investigated in future research.

## 5. Limitations

Data collection was conducted during COVID-19 restrictions in our institute. Therefore, all tests under exercising conditions were performed outdoors, where the standardisation of environmental issues (change in temperature, rain) was not possible. Performance diagnostics before the start of the study were not feasible as time spent in the institute needed to be as short as possible. Therefore, HR_max_, which we used in order to standardise the subject’s running intensity, was calculated using 220 minus age, which is generally considered to be inaccurate. As we consider newer concepts and equations to estimate HR_max_ to be limited, e.g., by small sample size and specific characteristics of the sample [47,48,49], we chose the most established equation by Fox et al. (1968) despite the criticism. Due to the focus of this study to compare paired ISF and CB data for estimating the application in practice, we rate this limitation to only have a low impact on results. For the herein-employed test (*t*-test with matched pairs), G*Power calculated a sample size of 15. As this sample size was eventually not reached, the effect sizes were calculated post hoc (described in 2.4 Statistical tests, ll.205–206), transparently demonstrating the power of the generated data. Due to the high effort of the study protocol (nine tests within 14 days) and a certain level of athleticism that was required (all subjects were following a training plan), only 10 subjects finished all tests. Although the originally aimed sample size was not achieved, the data are considered valuable, providing insights into the applicability of CGM in sports and, most importantly, representing a pilot study that serves as a basis for further research. Further, there is some criticism about statistical tests to evaluate CGM devices. As most concepts were developed for the evaluation of self-monitoring blood-glucose devices measuring in the same media as lab-method analyser [21,26,50,51], due to natural physiological differences, the applicability for CGM devices is debatable [12,52].

## 6. Conclusions

In summary, the results presented herein suggest that ISF glucose concentrations reflect CB glucose concentrations under resting conditions better than under exercising conditions in healthy subjects, which is in line with previous studies in subjects with diabetes. Consequently, using the sensor under exercising conditions as suggested by manufacturers, sensor-derived data could lead to misinterpretation by athletes. Therefore, it appears sensible to consult medical and/or nutritional experts if healthy athletes decide to utilise CGM. Practitioners should evaluate the risks and potential of the sensor considering the individual athlete.

## Figures and Tables

**Figure 1 ijerph-20-06440-f001:**
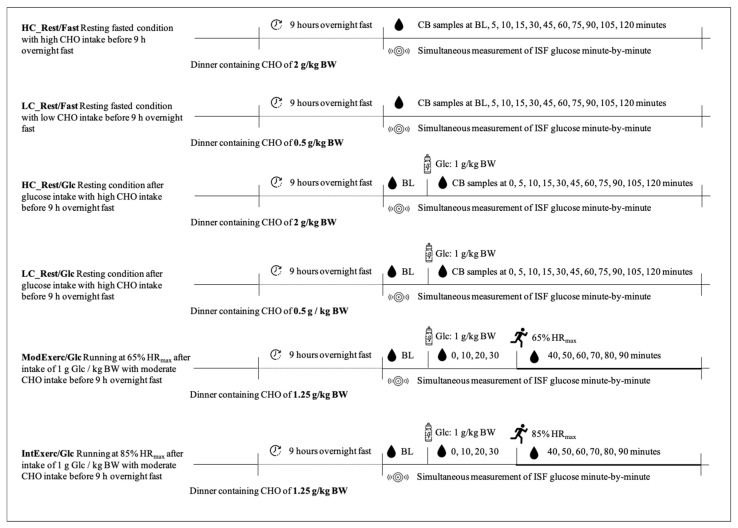
Fasted test protocol for resting conditions after high carbohydrate (CHO) dinner (HC_Rest/Fast) or a low CHO dinner (LC_Rest/Fast). Postprandial test protocol for resting conditions after a high CHO dinner (HC_Rest/Glc) or a low CHO dinner (LC_Rest/Glc). Postprandial test protocol for running at 65% HR_max_ or 85% HR_max_ after a moderate CHO dinner (ModExerc/Glc, IntExerc/Glc). BLOOD SPOT = CB sample collection, SENSOR = time of consideration of sensor-generated data, BOTTLE = ingestion of glucose solution containing 1 g Glc/kg BW, RUNNING PERSON = Start of activity.

**Figure 2 ijerph-20-06440-f002:**
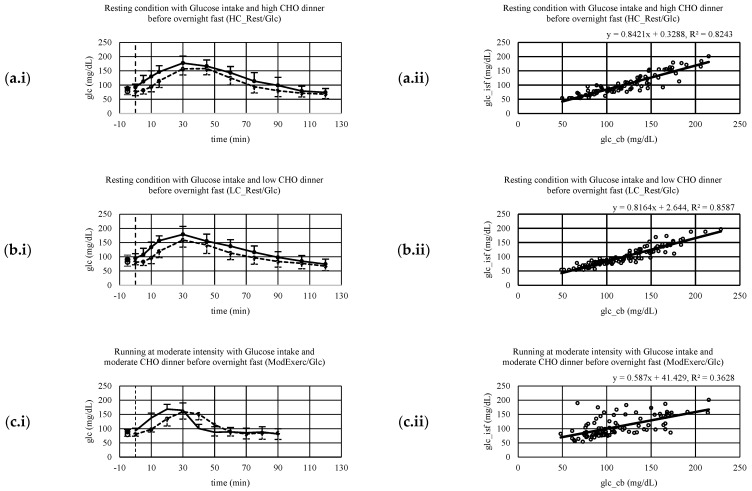
(**a.i**−**d.i**) Mean capillary blood (CB) and interstitial fluid (ISF) glucose concentrations including standard deviation (SD) (CB = solid line and positive error indication reflecting SD; ISF = dashed line and negative error indication reflecting SD). (**a.ii**−**d.ii**) Linear regression analysis of CB and ISF glucose concentration; spots reflecting paired individual data points. (**a**) results of high carbohydrate dinner (2 g/kg BW) before an overnight fast, that separated dinner and sample collection under resting conditions after an intake of a 1 g glucose/kg BW the next morning by at least 9 h (HC_Rest/Glc) (**b**) results of low carbohydrate dinner (<0.5 g/kg BW) before an overnight fast, that separated dinner and sample collection under resting conditions after an intake of a 1 g glucose/kg BW the next morning by at least 9 h (LC_Rest/Glc) (**c**) results of moderate carbohydrate dinner (1.25 g/kg BW) before an overnight fast, that separated dinner and sample collection under running for 60 min at a constant load of 65% of their maximal heart rate after an intake of a 1 g glucose/kg BW the next morning by at least 9 h (ModExerc/Glc), running started 30 min after glucose intake (**d**) the results of moderate carbohydrate dinner (1.25 g/kg BW) before an overnight fast, that separated dinner and sample collection under running for 60 min at a constant load of 85% of their maximal heart rate after an intake of a 1 g glucose/kg BW the next morning by at least 9 h (IntExerc/Glc), running started 30 min after glucose intake.

**Figure 3 ijerph-20-06440-f003:**
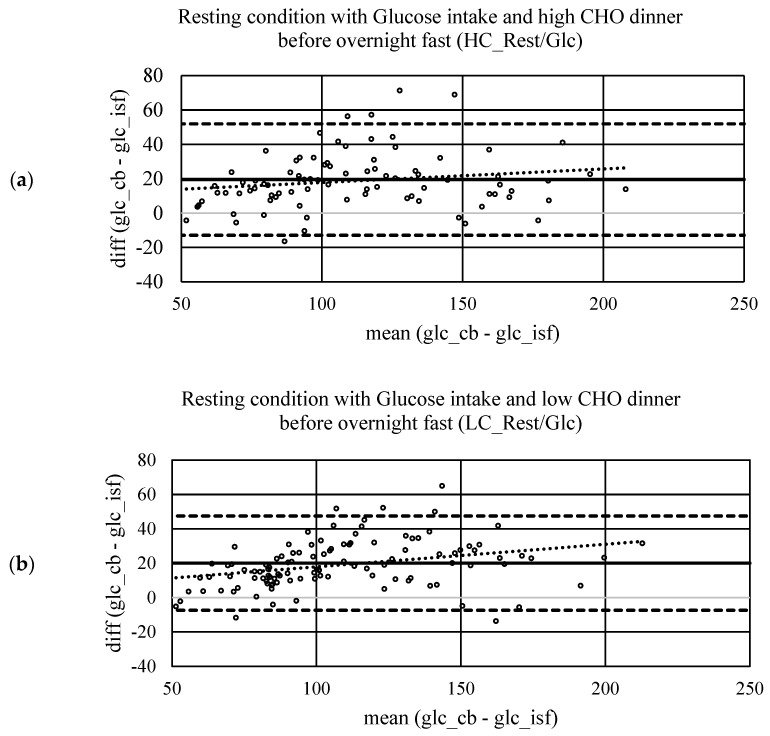
Systematic measurement difference of capillary blood (CB) and interstitial fluid (ISF) glucose concentration by Bland–Altman plot showing mean difference and their 95% confidence interval as lower and upper limit of agreement. (**a**) results of high carbohydrate dinner (2 g/kg BW) before an overnight fast, that separated dinner and sample collection under resting conditions after an intake of a 1 g glucose/kg BW the next morning by at least 9 h (HC_Rest/Glc) (**b**) results of low carbohydrate dinner (<0.5 g/kg BW) before an overnight fast, that separated dinner and sample collection under resting conditions after an intake of a 1 g glucose/kg BW the next morning by at least 9 h (LC_Rest/Glc) (**c**) results of moderate carbohydrate dinner (1.25 g/kg BW) before an overnight fast, that separated dinner and sample collection under running for 60 min at a constant load of 65% of their maximal heart rate after an intake of a 1 g glucose/kg BW the next morning by at least 9 h (ModExerc/Glc), running started 30 min after glucose intake (**d**) the results of moderate carbohydrate dinner (1.25 g/kg BW) before an overnight fast, that separated dinner and sample collection under running for 60 min at a constant load of 85% of their maximal heart rate after an intake of a 1 g glucose/kg BW the next morning by at least 9 h (IntExerc/Glc), running started 30 min after glucose intake.

**Figure 4 ijerph-20-06440-f004:**
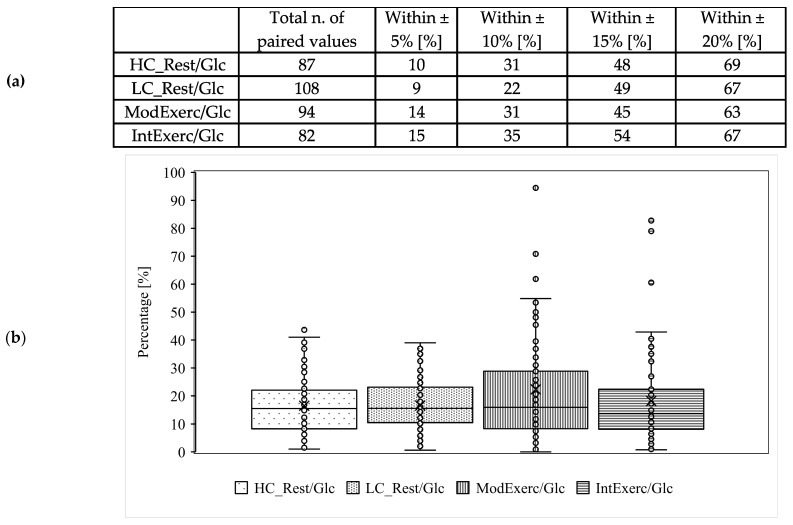
Dispersion of ARD data via calculation of percentages (**a**) within ranges of ±5, 10, 15 and 20% (**b**) and visualised as box whisker plot showing median ARD, lower and upper interquartile range, minimum and maximum values such as individual ARD values. Values differ from text, as figure shows distribution of MARD in individuals. HC_Rest/Glc: high carbohydrate dinner (2 g/kg BW) before an overnight fast, that separated dinner and sample collection under resting conditions after an intake of a 1 g glucose/kg BW the next morning by at least 9 h; LC_Rest/Glc: low carbohydrate dinner (<0.5 g/kg BW) before an overnight fast, that separated dinner and sample collection under resting conditions after an intake of a 1 g glucose/kg BW the next morning by at least 9 h; ModExerc/Glc: moderate carbohydrate dinner (1.25 g/kg BW) before an overnight fast, that separated dinner and sample collection under running for 60 min at a constant load of 65% of their maximal heart rate after an intake of a 1 g glucose/kg BW the next morning by at least 9 h, running started 30 min after glucose intake; IntExerc/Glc: moderate carbohydrate dinner (1.25 g/kg BW) before an overnight fast, that separated dinner and sample collection under running for 60 min at a constant load of 85% of their maximal heart rate after an intake of a 1 g glucose/kg BW the next morning by at least 9 h, running started 30 min after glucose intake.

**Table 1 ijerph-20-06440-t001:** Constitution of standardised meals that subjects had for dinner at least 9 h before their test. Meals were isocaloric but differed in macronutrient distribution in order to provide different amounts of carbohydrate. CHO = carbohydrate.

	High Carbohydrate Dinner	Moderate Carbohydrate Dinner	Low Carbohydrate Dinner
Energy (kcal/kg BW)	11.44	11.14	10.78
Carbohydrate (g/kg BW)	1.97	1.25	0.51
Protein (g/kg BW)	0.34	0.60	0.42
Fat (g/kg BW)	0.21	0.40	0.78

## Data Availability

Data are available upon request.

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
