# Peer review of "Continuous Glucose Monitoring (CGM) in Sports—A Comparison between a CGM Device and Lab-Based Glucose Analyser under Resting and Exercising Conditions in Athletes"

_ijerph, 2023, doi:10.3390/ijerph20156440_

Round 1
Reviewer 1 Report
First of all, I would like to thank the researchers for conducting this research. Glucose monitoring methods were compared in this study, the results of which can be used directly by trainers and athletes in the field. Consequently, in athletes, although under resting fasted conditions, both methods produce similar outcomes, under resting postprandial and exercising conditions, respectively, there are differences between methods. A few minor issue needs to be again handled which can be seen in pdf version as notes.

Author Response
Dear reviewer,
please see our response in the upload file.
Kind regards,
the authors

Reviewer 2 Report
I don't really understand what the purpose of this research was. the CGM used for people not diagnosed as I or II sugar is pointless. Why test something that is not required? CGM makes sense when the subject's body is unable to regulate blood glucose levels on its own. A healthy body in a situation of increased glucose supply produces more insulin in order to maintain normal blood glucose levels. In the event of a decrease in the level below the norm, insulin production is reduced, the feeling of hunger is turned on, and in extreme situations, glucagen is released.
The presented studies are not applicable in practice because there is no need to monitor the level of glucose in interstitial spaces in healthy people.
from the methodological point of view, the authors did not take into account the measurement shift that occurs in the case of CGM measurement in relation to blood measurement.
The results would be definitely more interesting if the athletes were not limited in nutrition, for example - in the case of this study, the authors provided a specific amount of glucose, a specific poor meal and what did they want to achieve by it?
In my opinion, the work does not bring any scientific value.
Author Response

(The authors gave the same response as above.)

Reviewer 3 Report
The research is design appropriate and the results are clearly presented, but the study group is too small. Please, add 10-20 athletes,. The results will be more reliable.
There are some mistakes. Please edit again.
Author Response

(The authors gave the same response as above.)

Reviewer 4 Report
The paper was written with high quality and scientific soundness. The authors described all the methods applied in this study and addressed the study's limitations. Although the study group had few patients, this is a pilot study.
The authors presented a pilot study of their research in which they analyzed if there are differences between two different glucose measurement methods in athletes performing different physical activities. It is not clear why these differences may count. The authors should explain what is the main study about and why these results could bring novelty to this domain. The authors also mentioned that this study was performed after an intake of 1g glucose/kg BW but should be given more details about how this glucose intake was administered. The topic is new but should motivate more clearly the aim of the study, explaining why measuring interstitial glucose could improve the performance of these athletes. I am not aware of other similar studies. The authors stated that the athletes performed six tests, but I counted 4. If I misunderstood, please correct me. The conclusions and references are good enough. Overall, I find the article well written, but the main issue remains convincing readers why using CGM is important in athletes without diabetes.
Author Response

(The authors gave the same response as above.)
